# Towards stable real-world equation discovery with assessing differentiating quality influence

**Mikhail Masliaev**
ITMO University
St. Petersburg, Russia, 197101
maslyaitis@gmail.com

**Ilya Markov**
ITMO University
St. Petersburg, Russia, 197101
iomarkov@itmo.ru

**Alexander Hvatov**
ITMO University
St. Petersburg, Russia, 197101
alex_hvatov@itmo.ru

## Abstract

This paper explores the critical role of differentiation approaches for data-driven differential equation discovery. Accurate derivatives of the input data are essential for reliable algorithmic operation, particularly in real-world scenarios where measurement quality is inevitably compromised. We propose alternatives to the commonly used finite differences-based method, notorious for its instability in the presence of noise, which can exacerbate random errors in the data. Our analysis covers four distinct methods: Savitzky-Golay filtering, spectral differentiation, smoothing based on artificial neural networks, and the regularization of derivative variation. We evaluate these methods in terms of applicability to problems, similar to the real ones, and their ability to ensure the convergence of equation discovery algorithms, providing valuable insights for robust modeling of real-world processes.

## 1 Introduction

Data-driven dynamical system modeling in forms of explicitly stated differential equations has emerged as a new direction for machine learning. Use of differential equations brings vast generalization ability, linked to the discovered expression interpretability and the existence of the equation general solution. The continuous process, represented by a multidimensional dataset, can be described with a partial differential equation (PDE), an expression connecting the dynamics along different.

The first advances in data-driven discovery of differential equations, as in [1], were devoted to the concept of applying symbolic regression to optimize the expression of the equation in a less restricted fashion. The next group of methods uses regularized regression with the help of the LASSO operator [2]. The expansive study has been carried out by the SINDy framework development team, examining the ability of the approach to derive ordinary and partial differential equations in work [3], and improving the expression quality obtained [4]. The evolution-based optimization approach, introduced and developed in works [5, 6, 7], involves creation of differential equations from elementary functions without assumptions about the structure of the equation. Although having reduced search space, the approach is able to mitigate the issue of infeasible candidate proposal and extreme computational costs, linked to the symbolic regression.

The majority of data-driven differential equation derivation techniques heavily rely on candidate libraries of numerically calculated derivatives of the data. We consider the implementation of

37th Conference on Neural Information Processing Systems (NeurIPS 2023).

noise-resistant approaches of differentiation as a vital sphere of development in the direction of increasing the applicability of equation-based methods to real-world problems. In such problems, the measurements are never taken with ideal accuracy. The commonly used approach to calculate the derivatives from the empirical data involves using a finite-difference schema. Despite being able to provide the noiseless data derivatives with decent quality, the finite differences amplify the noise in the input data, making high-order derivatives rather nondescriptive about the general dynamics of the governing process. This issue corresponds to the general ill-posed statement of the problem.

The problem of stable numerical differentiation has seen significant attention in the past years due to its importance in technical problems, apart from the differential equations discovery. For example, the denoising algorithms, which operate by solving the inverse problem of data reconstruction with derivatives, where the optimized functional is regularized by total variation of the data, were initially introduced in [8] for the purposes of image reconstruction and object edge detection, while high frequency filtering. This work is devoted to studying the applicability of stable differentiation methods to the discovery of data-driven equations.

## 2   Stable differentiation problem statement and proposed methods

In what follows, we will primarily discuss the reduction of random error in the measurements. While other sources of inaccuracies in the data, such as systematic errors, can be significant, they tend to be elusive even though significantly affecting the resulting data-driven equation.

Let us denote the input data for the differential equation discovery algorithm as $u(t, \mathbf{x})$, which is collected as measurements and, in addition to the correct state of the system $\overline{u}(t, \mathbf{x})$ contains noise $n(t, \mathbf{x})$. While we can be sure in the presence of noise in the data, several assumptions can be made about the distribution $F$, to which it belongs. The measurement at the point $(t, \mathbf{x})$ is assumed to be drawn from the Gaussian distribution (Additive White Gaussian Noise, AWGN) with its mean $\overline{u}(t, \mathbf{x})$ - the correct value of the underlying process. In our experiments we introduce the noise standard deviation $\sigma$ dependent on the variable state $\sigma = \kappa \overline{u}(t, \mathbf{x})$.

$$u(t, \mathbf{x}) = \overline{u}(t, \mathbf{x}) + n(t, \mathbf{x}), \ n(t, \mathbf{x}) \sim F(t, \mathbf{x}) \tag{1}$$

This section is devoted to the presentation of alternative tools for calculating the derivatives of a modeled function. The baseline approach to the numerical differentiation involves finite-difference schema that employ values of the dependent variable in grid nodes in order to calculate its derivatives.

$$\frac{\partial u(t, \mathbf{x})}{\partial x_i} \approx \frac{\Delta_{\delta, i} u}{\delta_i} = \frac{u(t, \mathbf{x} + \delta_i) - u(t, \mathbf{x})}{\delta_i}, \tag{2}$$

where the partial derivative of the data-representing function $u(t, \mathbf{x})$ over the $i$-th spatial axis is reconstructed with the values in nodes $(t, \mathbf{x} + \delta_{\mathbf{i}})$ and $(t, \mathbf{x})$ with the finite-difference operator "forward" $\Delta_{\delta, i}$. By $\delta_{\mathbf{i}}$ we denote the vector of increment over the $i$-th axis, $\delta_i^j = 0, i \neq j$, and $\delta_i^i$ - non-zero step of the grid.

With the data contaminated in the manner presented in Eq. 1, it is possible to estimate the quality of derivatives, based on the finite differences. Let us assume that the input data on the compact $\Omega$ belong to the Sobolev space $W^{k,p}(\Omega)$ of functions that have their derivatives up to $k$-th order belong to the $L^p(\Omega)$ space (have finite Lebesgue integral): $\overline{u} \in W^{k,p}(\Omega)$. Although we cannot be sure of the same properties of the observation $u$, it can still be attributed to the Lebesgue space with $\infty$-norm: $u \in L^\infty(\Omega)$. In this case, the finite-difference discrepancy can be estimated from the norms in the corresponding spaces:

$$\|\overline{u}'_{x_i} - \frac{\Delta_{\delta, i} u}{2\delta_i}\|_p \leq \|\frac{\Delta_{\delta, i}(u - \overline{u})}{2\delta_i}\|_p + \|\overline{u}'_{x_i} - \frac{\Delta_{\delta, i} \overline{u}}{2\delta_i}\|_p \leq \frac{2\delta_i}{h} + \frac{hC}{2}, \tag{3}$$

where by $\|\cdot\|_p$ we denote the norm in space $L^p(\Omega)$, and $C$ is the constant obtained from the Taylor series derivation of finite differences: $C \geq \|f''_{x_i x_i}\|_p$. This estimation indicates that the derivatives are sensitive to the errors in the measurement. Furthermore, the reduction of the grid step, which

is usually preferable due to the lower pure numerical error in the finite difference, leads to the magnification of random errors.

The noise influence on the data can be viewed from the point of view of Fourier analysis. The studied process shall not produce high-frequency oscillations, or have amplitudes significantly lower than the low-frequency counterparts. If the opposite is true, the data may have aliasing problems, thus limiting the applicability of the frequency-based analysis. We can note that these high-frequency components in the DFT (discrete Fourier transform) are linked to the measurement noise or small-scale processes that shall be omitted during the equation construction, and shall be filtered out. In what follows, brief notes of applied differentiation methods are presented, with a more detailed and expanded formulation placed in the Appendix A.

- **Filtering-based approaches:** One of the approaches considered in this work involves approximation of the input data with the fully connected artificial neural network (ANN). One of the valuable properties of the artificial neural network is that the low-frequency signal in the data is learned first, while further training approximates the high-frequency components [9]. Thus, by training an ANN representation of the process, we can obtain its low-frequency approximation, which can be further differentiated with decreased noise component.

  Savitzky-Golay (SG) filtering, developed in [10], is a commonly used approach to signal or data filtering, coupled with an opportunity to compute derivatives, involves a least squares-based local fitting of the polynomials to represent the data. For each grid node, the data in its proximity is used to construct a polynomial that can be analytically differentiated.

- **Spectral domain differentiation:** Although the process of differentiation in the spatial domain can be complicated for the data, described with an arbitrary function, in the Fourier domain the derivatives can be estimated on a term-to-term basis [11]. The discrete Fourier transform (DFT) is the basis of our implementation of spectral domain differentiation. In the spectral domain, integration and differentiation can be maintained by multiplication of series terms with an appropriate exponential. This leads to low computational costs, especially if the data are located on the uniform grid, thus allowing use of the Fast Fourier Transform instead of DFT. The signal filtering is done with the Butterworth filter that is able to preserve signal with frequencies lower than the cutoff frequency, while dampening the high-frequency ones.

- **Total variation regularization:** Variational principles provide an alternative method that incorporates inverse problem solution with the regularization of the gradient variation, or its higher-order analogues (e.g. Hessian). One of the main advances in this field was made in [12, 13].

## 3 Experiment section

The investigation and validation of theoretical approaches proposed in the previous sections is done with a comprehensive numerical experiment. Experiments are performed with two of the most commonly employed approaches for discovering data-driven differential equations: sparse regression and evolutionary-based approaches. The lack of analytical solutions for both problems necessitates a study of the algorithms' behavior.

### 3.1 Sensitivity of the LASSO operator based approach

To evaluate the benefits of implementing stable differentiation on the quality of differential equations, discovered by LASSO regression, we have conducted a series of experiments with the SINDy framework. To analyze how the selection of the differentiation method affects the coefficients of the equation, we have conducted a series of experiments on the solution of a linear system $x' = ax + by$, $y' = cx + dy$, with $a = -0.1$, $b = 2$, $c = -2$, and $d = -0.1$, provided by the developers of the SINDy framework. The noise was added only to the data to differentiate, leaving the candidate library intact. Otherwise, the results will be primarily influenced by the smoothing module, not the stable differentiation.

The summary of the experiment is presented in Fig. 1, where the performance of the main differentiation methods was compared in 25 independent runs for each noise level. Notably, experiments

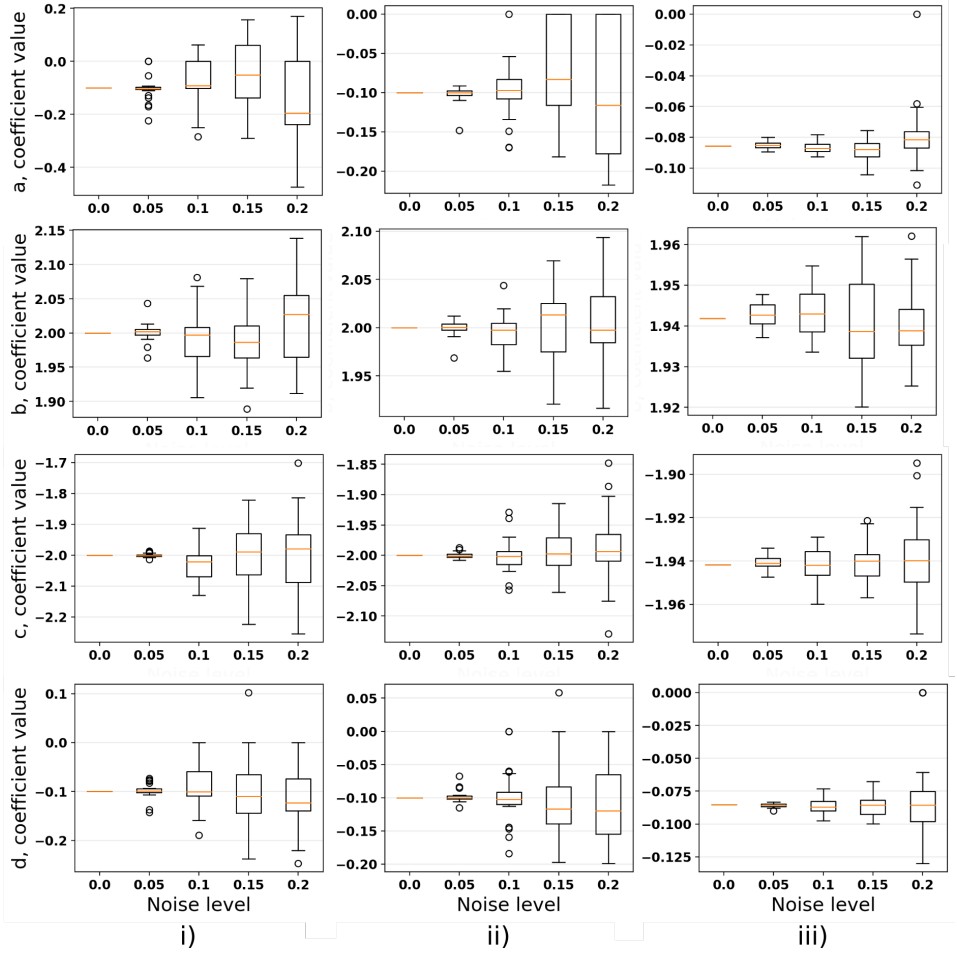

Figure 1: Statistics of coefficients of the equations, obtained by sparse regression with different differentiation approaches: i) finite-difference schema, ii) Savitzky-Golay filtering, iii) spectral method. The noise level $\kappa$ denotes scale $\sigma_x = \kappa * x(t)$, $\sigma_x = \kappa * x(t)$ of the Gaussian distribution, from which the random errors are sampled

that employ variation regularization have not produced decent equations. The method constructs an approximation of the derivative close to the broken line. While it may be sufficient in problems of contour recognition on images, a more nuanced approach, which can preserve the structure of derivatives, is necessary in equation discovery. While all three compared methods correctly operate on noiseless data (with the spectral method introducing minor bias due to low number of non-dampened frequencies), on the corrupted datasets the spectral method has the highest stability.

## 3.2 Experiments on sensitivity of the evolutionary approach

To better understand the effects of stable differentiation on the structures obtained by evolutionary algorithm equations, we conducted a series of experiments on synthetic data. All data for these experiments were obtained from the solutions of a priori known equations, as in the previous set of experiments, making the validation of the equation search explicit. As the metric of the equation search correctness, we employ the fitness function values and a proportion of the equations with desired structures among the individuals on the Pareto-optimal set of equations.

To provide some diversity among the problem statements, we have performed a data-driven rediscovery of the following differential equations: an ordinary differential equation $mu'' + qu' + kx = 0$ with parameters $m = 1$, $c = 0.25$, and $k = 3$, and the wave equation $u''_{tt} = c^2 u''_{xx}$, $c = 0.5$. In

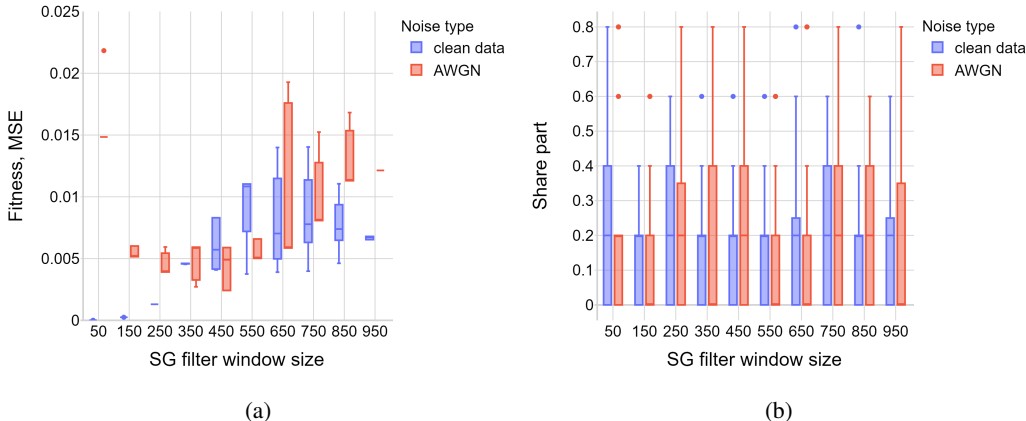

(a)             (b)

Figure 2: Results of the ordinary differential equation discovery experiment on clean data and AWGN-noised dataset with $\kappa = 0.1$ with different window size of Savitzky-Golay filter. Frame a) indicate the process representation error of the obtained equations, and b) shows the prevalence of equation with correct structures on Pareto-optimal set.

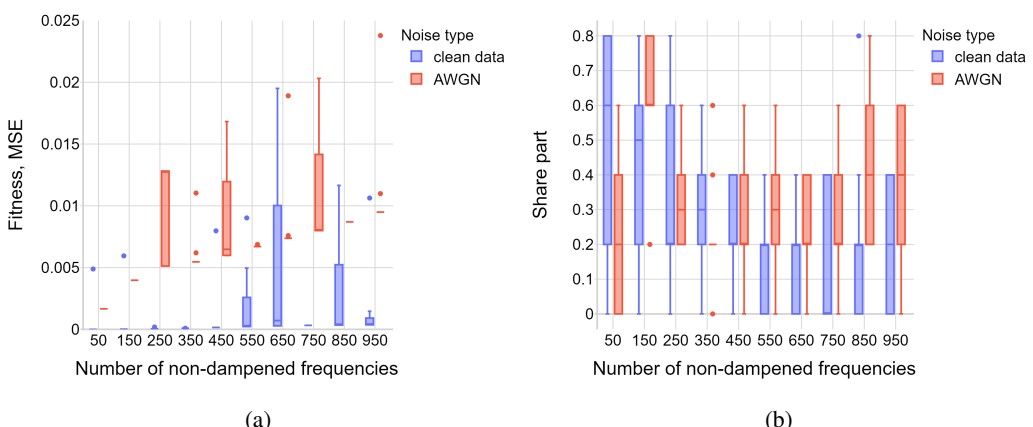

(a)             (b)

Figure 3: Results of the ordinary differential equation discovery experiment on clean data and AWGN-noised dataset with $\kappa = 0.1$ with different number of frequencies, left to be unchanged by the Butterworth filter. Frame a) indicates the process representation error of the obtained equations, and b) show the prevalence of equation with correct structures on Pareto-optimal set.

contrast to the sparsity-promoting methods, evolutionary algorithms can distil differential equations of higher orders in explicit form, i.e. not by translating them to a differential equation of higher order.

For the experiments, we have selected three different methods for obtaining numerical derivatives from input data: finite-differences, calculated based on the ANN approximation of input data; spectral differentiation and Savitzky-Golay filtering. Due to the sensitivity of the aforementioned methods to the parameters, a series of equation searches were conducted to better understand the bounds of the equation search errors.

To investigate the impact of noise in real data on the evolutionary algorithm, normally distributed noise with the following characteristics was added to the data. To take into consideration the stochastic nature of evolutionary optimization, multiple optimization runs were conducted while preserving the numerical differentiation parameters.

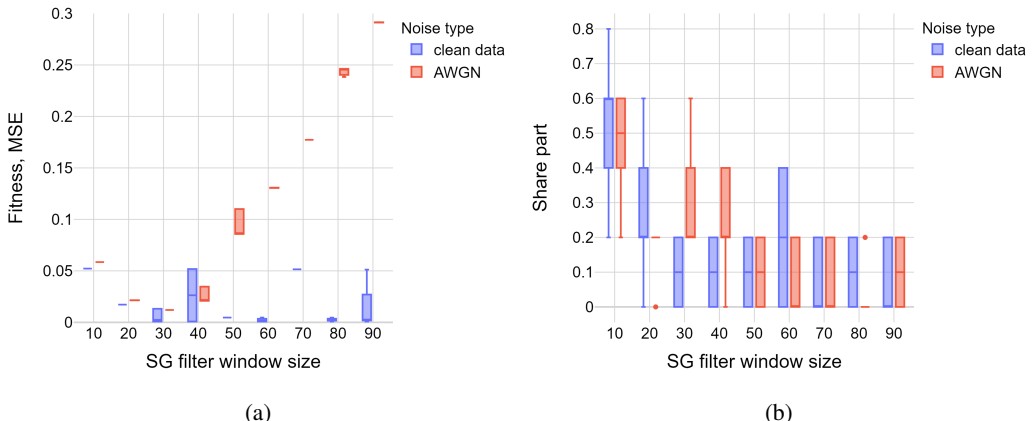

Figure 4: Results of the wave equation discovery experiment on clean data and AWGN-noised dataset with $\kappa = 0.1$ with different window size of Savitzky-Golay filter. Frame a) indicate the process representation error of the obtained equations, and b) shows the prevalence of equation with correct structures on Pareto-optimal set.

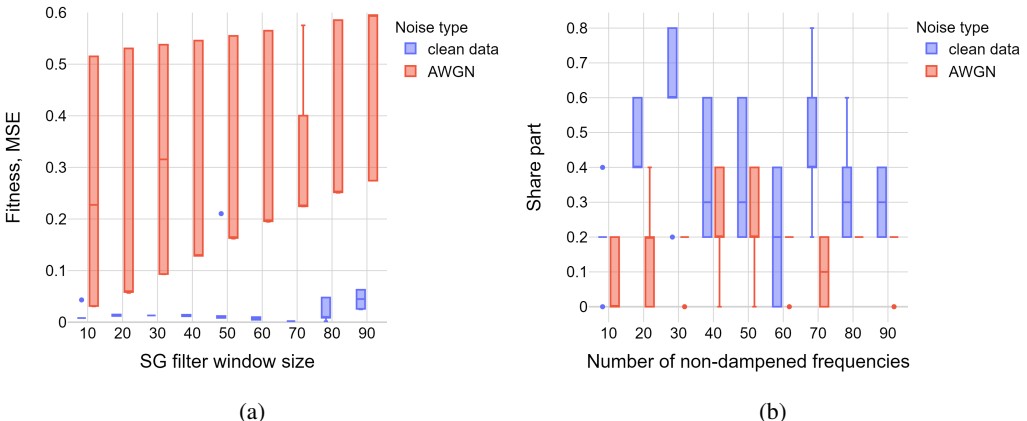

Figure 5: Results of the wave equation discovery experiment on clean data and AWGN-noised dataset with $\kappa = 0.1$ with different number of frequencies, left to be unchanged by the Butterworth filter. Frame a) indicates the process representation error of the obtained equations, and b) show the prevalence of equation with correct structures on Pareto-optimal set.

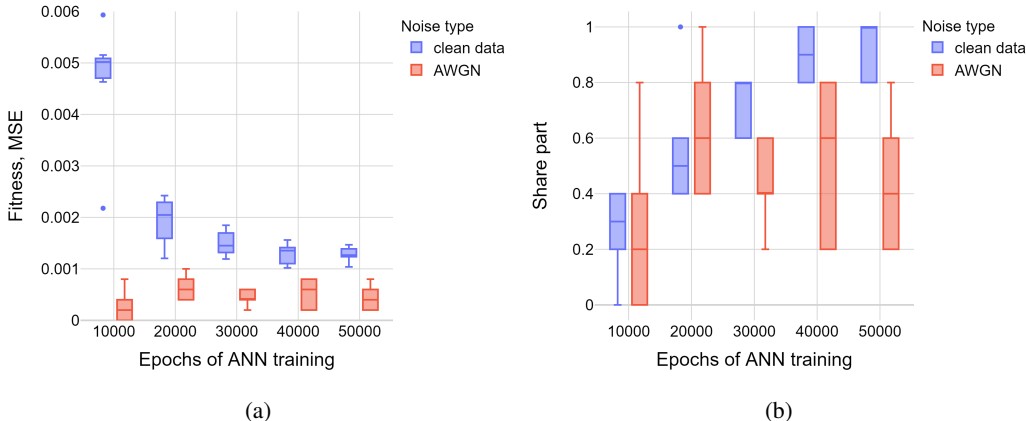

(a)                                                    (b)

Figure 6: Results of the ordinary differential equation discovery experiment on clean data and AWGN-noised dataset with $\kappa = 0.1$ with different window size of Savitzky-Golay filter. Frame a) indicate the process representation error of the obtained equations, and b) shows the prevalence of equation with correct structures on Pareto-optimal set.

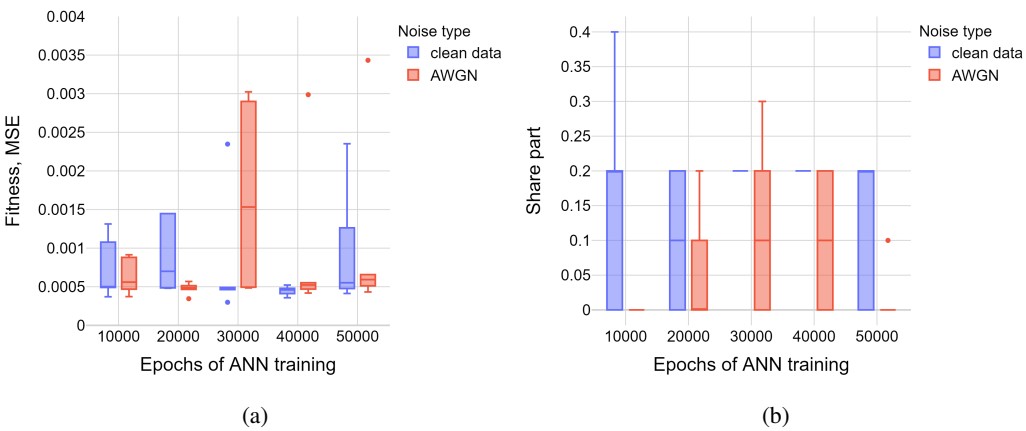

(a)                                                    (b)

Figure 7: Results of the wave equation discovery experiment on clean data and AWGN-noised dataset with $\kappa = 0.1$ with different epochs of approximating artificial neural network training. Frame a) indicates the process representation error of the obtained equations, and b) show the prevalence of equation with correct structures on Pareto-optimal set.

The results of experiments on evolutionary differential equation discovery, presented on Fig. 2, Fig. 3, Fig. 4, and Fig. 5, indicate, that an increase in differentiation error leads to an escalation in final model errors and a reduction in the proportion of equations with the correct structure in the Pareto frontier. The noise added to the data increases the dispersion of model errors, but still maintains the trend outlined for clean data.

The behavior of algorithms on ANN-filtered data, presented in Fig. 6 and Fig. 7 shows tendencies that differs from stated above. The differentiation error decreases and stabilizes for the derivatives in partial differential equations discovery. Due to stochastic behavior of neural network learning process, this effect leads to diminishing of model errors and increase in share part of equations with right structure in the final evolution optimization epoch. The same effects occur in partial differential equation discovery. Despite the decrease in derivation error, portrayed on Fig. 8, it may not stabilize if the data are highly contaminated. This leads to a high variance of model errors and almost eliminates correct equation structures from the final Pareto set when noise is added.

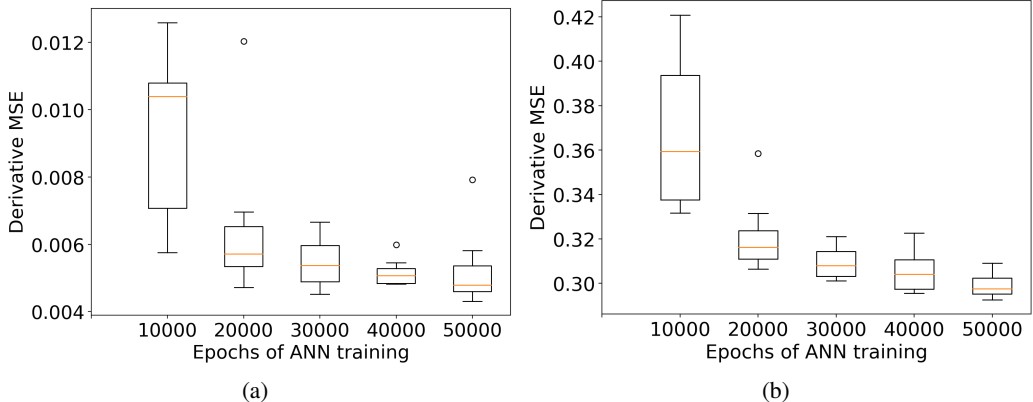

Figure 8: Errors of the derivative calculations, based on the wave equation solution, using artificial neural networks with different epochs number, a) portrays algorithm behavior on clean data, while b) is related to the added Gaussian noise.

## 4 Conclusion

The ability to train models in form of differential equations for the real-world processes is the next step in the development of the equation discovery techniques. Although some works state that it is achieved, most of the papers still consider toy examples with a known solution and known a priori equation form. During the experiments, following points could be outlined that will define the step of both gradient LASSO methods and evolutionary approaches:

- For the real-world equation discovery, as the experiments show, it is crucial to choice proper differentiation method rather than discovery method by itself;

- The filtering could not be applied to achieve arbitrary smoothness, since we need to preserve the information to restore the process;

- In real-world applications we have to somehow deal with the pre-defined library in cases when the underlying process known at a very high scale, i.e. we know origin of data, but not the equation.

The current study introduces one more control variable to make the discovery of real-world equations more viable. Namely, before choosing the method of discovery, we have to differentiate noisy experimental data. Even in the considered toy examples, it is still a challenge for existing state-of-the-art differentiation method to be able to both handle noise and recover the correct equation structure of the equation. The classical finite difference method is not good enough, instead we have to use more advances differentiation techniques - filtration, neural network approximation, or regularization for every problem appearing. For the real world equation learning scenarios we propose to use stable differentiation in all studies: when the random noise is assumed to have high magnitudes, spectral differentiation or ANN-based filtering are preferable, while in cases of low data distortions Savitsky-Golay filtering may be enough for decent equation discovery.

## 5 Data and code availability

The experiments are available in the repository `https://anonymous.4open.science/r/ai4science_stable_diff_exp-735B/`.

## Acknowledgments and Disclosure of Funding

This work was supported by the Analytical Center for the Government of the Russian Federation (IGK 000000D730321P5Q0002), agreement No. 70-2021-00141.

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

# A Differentiation approach formulation

## A.1 Savitzky-Golay filtering

Savitzky-Golay (SG) filtering, developed in [10], is a commonly used approach to signal or data filtering, coupled with an opportunity to compute derivatives, involves a least squares-based local fitting of the polynomials to represent the data. To the set of data samples along an axis, we introduce the window of (commonly, odd) length $N = 2M + 1$, allowing the construction of series of polynomials $P_0(x), P_1(x),$ ... up to (even) order $n, n < N$ to approximate the data in the interior of our domain. With the selection of appropriate window size, from which the function values are used for the approximation, and polynomial order, the overdetermined system is constructed. Its solution provides the polynomial coefficients that represent the smoothed signal, without oscillations, caused by the random error. Even though the boundaries of length $M$ can be processed in a separate way, with the finite-difference schema or by a shifted approximation, the quality of results tend to decrease, thus for the equation discovery only the domain interior shall be used.

During the calculation of the partial derivative $u'_j$ for the sample $u(x_i)$, matching the $x_i$ grid node along the $j$-th axis, we select samples $\mathbf{u}_i = (u_{i-M}, u_{i-M+1}, \ ... \ , u_i, \ ... \ , u_{i+M})$ in the aforementioned window. Using the corresponding coordinates $\mathbf{y}_i = (x_{i-M}, \ ... \ , x_i, \ ... \ , x_{i+M})$, we introduce the least-square problem of detecting coefficient vector $\alpha = (\alpha_0, \ ... \ , \alpha_{n-1})$ for the series $P_0, \ ... \ , P_{n-1}$. The representation of data samples is as follows:

$$u_i = \sum_{k=0}^{n-1} \alpha_k P_k(x_i). \tag{4}$$

$$\alpha = \arg\min_{\alpha'} |\mathbf{u}_i - P\mathbf{y}_i|, \tag{5}$$

where matrix $P$ contains values of the polynomials in the grid nodes.

In our case, we utilize orthogonal Chebyshev polynomials of the first kind, where by $C_m^{2k}$ we denote the number of combination of $2k$ elements from the set of cardinality $m$:

$$T_m(x) = \sum_{k=0}^{\lfloor m/2 \rfloor} C_m^{2k}(x^2 - 1)^k x^{m-2k} \tag{6}$$

Having a series of Chebyshev polynomials with calculated coefficients, differentiation can be held analytically. Using the representation of data as series in 4, we get the derivative as $u'_i = \sum_{k=0}^{n-1} \alpha_k U_k(x_i)$, where $U_k$ is a Chebyshev polynomial of the second kind.

$$U_m(x) = \sum_{k=0}^{\lfloor m/2 \rfloor} C_{m+1}^{2k+1}(x^2 - 1)^k x^{m-2k} \tag{7}$$

Although the provided approach is capable of filtering the data and stably calculating the derivatives, work [14] suggests that modification of Savitzky-Golay filtering by adding fitting weights or by implementing other filters, such as Whittaker-Henderson filter, can lead to better results in noise suppression.

## A.2 Spectral domain differentiation

Although the process of differentiation in the spatial domain can be complicated for the data, described with an arbitrary function, in the Fourier domain the derivatives can be estimated in term-to-term basis [11]. In general, the series of the derivatives, taken on a term-to-term basis may not converge. However, if we assume that the data represents continuous piecewise smooth function that has piecewise differentiable derivatives, the data can be differentiated term-to-term.

A discrete Fourier transform (DFT) is the basis for our implementation of spectral domain differentiation. Let us examine a case of one-dimensional data, even though the algorithm can operate

on multi-dimensional data, with the canonical discrete Fourier transform algorithm replaced by n-dimensional DFT. In data-driven equation discovery problems, one-dimensional data $u(t)$ is viewed from the point of view of samples $u_n = u(nT/N), n = 0, 1, \ldots, N-1$, where $T$ is the length of time interval and $N$ - the number of samples, and the corresponding coordinates will be $t_n = nT/N, n = 0, 1, \ldots, N-1$. The Fourier coefficients are denoted as $\hat{u}_k$, and they are calculated as:

$$\hat{u}_k = \frac{1}{N} \sum_{n=0}^{N-1} u_n exp(-2\pi i \frac{nk}{N}). \tag{8}$$

In many cases, the data are provided on the regular (even multi-dimensional) grid, thus to improve the algorithm performance a fast Fourier transform can be used. Due to the lower computational complexity, the increase in performance is substantial. The process of data reconstruction, using the obtained Fourier coefficients, is held with an inverse discrete Fourier transform:

$$u_n = \sum_{k=0}^{N-1} \hat{u}_k exp(2\pi i \frac{nk}{N}). \tag{9}$$

Full term-by-term differentiation is performed in the Fourier domain, and the derivatives values are computed by the inverse DFT. For example, an expression for the first-order derivative has form, as in Eq. 10.

$$u'(t_k) = \sum_{0 < k < \frac{N-1}{2}} \frac{2\pi i}{T} k \left( \hat{u}_n exp(2\pi i \frac{nk}{N}) - \hat{u}_{N-k} exp(-2\pi i \frac{nk}{N}) \right). \tag{10}$$

Filtering with the desired properties can be done with low-pass filters that pass signals with lower frequencies, while dampen the high-frequency ones. Butterworth filter is a representative of such tools, and is flat for the passband (the frequencies that we do not want to penalize). The latter property prevents distortion of the modeled process by introducing factors, close to 1, to the low-frequency Fourier components. The penalizing factor is introduced with the expression eq. 11:

$$G(\omega) = \frac{1}{1 + (\omega/\omega_{cutoff})^{2s}}, \tag{11}$$

where $\omega$ is the frequency, $\omega_{cutoff}$ is the cutoff frequency, indicating the boundary frequency, from which the damping begins, and $s$ is the filter steepness parameter. The resulting expression is obtained with the introduction of penalizing factors $G(\omega) = G(k/N)$ into the series, representing derivatives:

$$u'(t_k) = \sum_{0 < k < \frac{N-1}{2}} G(k/N) \frac{2\pi i}{T} k \left( \hat{u}_n exp(2\pi i \frac{nk}{N}) - \hat{u}_{N-k} exp(-2\pi i \frac{nk}{N}) \right) \tag{12}$$

The derivative of the higher orders can be calculated recursively from the lower order ones with the same filtering-based differentiation procedures, or, preferably, by the further multiplication with the integrating coefficient and IDFT.

### A.3 Total variation regularization

Variational principles provide an alternative method that incorporates inverse problem solution with the regularization of the variation of the gradient or its higher order analogues (e.g. Hessian). Rudin-Osher-Fatemi model [8] in its discrete formulation can be represented by the optimization problem of minimizing functional 13.

$$|D(\nabla \cdot u)|_1 + \frac{\mu}{2}|K(\nabla \cdot u) - u|_2^2 \longrightarrow \min_u, \tag{13}$$

where $\nabla \cdot u = (\frac{\partial u}{\partial t}, \frac{\partial u}{\partial x_1}, \; ...)$ is the gradient of the data field and $K$ and $D = (D_t, D_{x_1}, D_{x_2}, \; ...)$ represent discrete integration operators onf differentiation. Regularization of gradient variation is maintained with term $|D(\nabla \cdot u)|_1 = \sum_\Omega \sqrt{\sum_{i,\,j} \frac{\partial^2 u)}{\partial x_i \partial x_j}}$.

[12, 13]

Although there are multiple approaches to the solution of the problem, we employ an approach, proposed in articles [12, 13], that is designed for a function of one variable. While this approach can be generalized to the problems of higher dimensionality, the computational costs associated with the optimization limit the method's applicability to large datasets. To perform the functional optimization required in Eq. 13, the corresponding Euler-Lagrange equation has to be formed and solved.

