# OpenReview forum: "Towards stable real-world equation discovery with assessing differentiating quality influence"
_NeurIPS.cc/2023/Workshop/AI4Science — NeurIPS2023-AI4Science Poster_

### Official Review · Reviewer_r54E · 2023-10-23
**Insights into differentiation approaches for data-driven differential equation discovery but lacking depth in practical applications and unique contributions**

**Rating:** 6
**Confidence:** 2

**Review:**

The paper investigates differentiation techniques for data-driven differential equation discovery, offering an analysis of four differentiation methods: Savitzky-Golay filtering, spectral differentiation, smoothing based on artificial neural networks, and the regularization of derivative variation. Through numerical experiments, the work establishes the importance of differentiation methods in equation discovery. However, a more detailed analysis of the results and their implications for real-world applications would further strengthen the paper. Clarification of the unique contributions of this work relative to existing literature would be beneficial.

---

### Meta-Review · Area_Chair_FqV7 · 2023-10-28

**Recommendation:** Accept (Poster)
**Confidence:** 2

**Metareview:**

The paper performs experiments to evaluate the quality of several existing auto-differentiation algorithms in lieu of finite differences to examine their stability in recovering coefficient values. Upon reading the paper, I agree that while the methodology examines quantitative aspects of stability under fairly narrow scenarios eg. LASSO, ODEs and the wave equation under idealized noise scenarios (eg normal distribution of noise). As alluded to by the reviewer, it is unclear what the main takeaway is in the context of real world applications (where often the underlying equation governing the function transformation could be highly complex, and the noise form unknown). It is not clear which of the examined methods is to be considered for real applications.